# Mixed methods approach to examining the implementation experience of a phone-based survey for a SARS-CoV-2 test-negative case-control study in California

Nozomi Fukui[1,2☯], Sophia S. Li[1,3☯], Jennifer DeGuzman[1], Jennifer F. Myers[1], John Openshaw[1], Anjali Sharma[4,5], James Watt[1], Joseph A. Lewnard[2,6,7], Seema Jain[1], Kristin L. Andrejko[1,3‡]*, Jake M. Pry[1,8‡]*, on behalf of the California COVID-19 Case-Control Study Team[1¶]

1 California Department of Public Health, Richmond, CA, United States of America, 2 Division of Epidemiology and Biostatistics, School of Public Health, University of California, Berkeley, CA, United States of America, 3 College of Agricultural and Environmental Sciences, University of California, Davis, CA, United States of America, 4 University of Washington, Hans Rosling Center, Global Health, Seattle, WA, United States of America, 5 The Centre for Infectious Disease Research in Zambia (CIDRZ), Lusaka, Zambia, 6 Division of Infectious Diseases & Vaccinology, School of Public Health, University of California, Berkeley, California, United States of America, 7 Center for Computational Biology, College of Engineering, University of California, Berkeley, CA, United States of America, 8 Department of Public Health Sciences, School of Medicine, University of California, Davis, CA, United States of America

☯ These authors contributed equally to this work.
‡ JMP and KLA are joint senior authors and contributed equally to this work
¶ Membership of the California COVID-19 Case-Control Study Team is provided in the Acknowledgments
* jmpry@ucdavis.edu (JMP); onp9@cdc.gov (KLA)

**Data Availability Statement:** De-identified data and supporting data documentation will be made

## Abstract

### Objective

To describe the implementation of a test-negative design case-control study in California during the Coronavirus Disease 2019 (COVID-19) pandemic.

### Study design

Test-negative case-control study

### Methods

Between February 24, 2021 - February 24, 2022, a team of 34 interviewers called 38,470 Californians, enrolling 1,885 that tested positive for SARS-CoV-2 (cases) and 1,871 testing negative for SARS-CoV-2 (controls) for 20-minute telephone survey. We estimated adjusted odds ratios for answering the phone and consenting to participate using mixed effects logistic regression. We used a web-based anonymous survey to compile interviewer experiences.

available through GitHub upon publication: https://github.com/noz-o-mi/CA-COVID-Case-Control-implementation.

**Funding:** This study was supported by the Centers for Disease Control and Prevention, Enhanced Epidemiology and Laboratory Capacity (ELC) grant number: 5-NU50CK000539. The funders had no role in study design, data collection and analysis, decision to publish, or preparation of the manuscript.

**Competing interests:** The authors declare the following financial interests/personal relationships which may be considered as potential competing interests: Joseph Leonard has received grants and honoraria from Pfizer, Inc, outside the submitted work. There are no patents, products in development or marketed products associated with this research to declare. This does not alter our adherence to PLOS ONE policies on sharing data and materials.

## Results

Cases had 1.29-fold (95% CI: 1.24–1.35) higher adjusted odds of answering the phone and 1.69-fold (1.56–1.83) higher adjusted odds of consenting to participate compared to controls. Calls placed from 4pm to 6pm had the highest adjusted odds of being answered. Some interviewers experienced mental wellness challenges interacting with participants with physical (e.g., food, shelter, etc.) and emotional (e.g., grief counseling) needs, and enduring verbal harassment from individuals called.

## Conclusions

Calls placed during afternoon hours may optimize response rate when enrolling controls to a case-control study during a public health emergency response. Proactive check-ins and continual collection of interviewer experience(s) and may help maintain mental wellbeing of investigation workforce. Remaining adaptive to the dynamic needs of the investigation team is critical to a successful study, especially in emergent public health crises, like that represented by the COVID-19 pandemic.

## Introduction

The Coronavirus Disease 2019 (COVID-19) pandemic induced rapid mobilization of public health research to inform policy [1]. Observational studies have played critical roles in defining COVID-19 epidemiology by identifying risk factors for infection and estimating the effectiveness of mitigation strategies [2–7]. Many observational studies conducted during the pandemic utilized remote technologies, such as phones, to safely enroll participants, however these platforms may pose unique challenges [8–12]. Understanding phone-based participation patterns throughout the pandemic may help optimize the implementation of future epidemiologic studies.

Prior to the pandemic, participation in phone surveys varied by disease, age, and time of day [11,13–15]. Individuals or individuals adjacent to person(s) who have history of disease are more likely to participate than unaffected individuals [11]. Younger people may be more willing to answer an unknown caller, but less willing to participate in a public health survey that involves disclosing sensitive information such as their recent contacts [14]. Additionally, the time of day that individuals are called may also influence participation [15]. Polarization of public health throughout the pandemic, including increasingly negative attitudes towards contact tracing, may limit willingness to participate in phone-based COVID-19 research [16–19]. In the novel, dynamic context of the pandemic, identification of predictors of participation in observational studies using remote technologies are limited. Public health professionals report substantial mental health burdens during the pandemic, yet details regarding the toll of sensitive research on researchers is scant [20–23].

We describe the implementation of a phone-based, test-negative SARS-CoV-2 case-control study in California during the COVID-19 pandemic. We estimate predictors of answering the phone, enrolling in the study, and identify reasons for refusing participation. Furthermore, we provide qualitative descriptions of interviewer experiences to identify successes and gaps in staff support systems. These components are critical to successful implementation and can inform future epidemiologic studies conducted throughout similar pandemic settings.

## Materials and methods

### Study design and enrollment

We reviewed data collected from February 24, 2021, to February 24, 2022, by the California Department of Public Health (CDPH) test-negative case-control study that evaluated risk factors for SARS-CoV-2 infection (**S1 File**) [6,7]. Potential case and control participants were defined as individuals with a positive and negative laboratory-confirmed SARS-CoV-2 test result, respectively. Cases and controls were individually matched by age group, sex, multi-county region, and test result window (≤7-day difference). Throughout the study period, trained interviewers used soft-phone technology with a California area code to call and facilitate a 20-minute survey in English or Spanish (**S1 File**). A script accompanied the electronic survey to standardize the participant experience (**S1 File**). Potential participants were informed of the 20-minute survey length before consenting.

Individuals were eligible to participate if they reported no clinical diagnosis of COVID-19 or positive test result for SARS-CoV-2 infection prior to their most recent test result. From January 6, 2022, as at-home test use increased, those with a previous (< 2 days) positive at-home test result became eligible. If not capable of answering questions, recruitment proceeded if a proxy respondent was available, and the potential participant gave informed consent both to participation and to have the proxy answer on their behalf.

Interviewers enrolled a case, followed by calls to 30+ matched controls, in a repeated case-control pair format. If unsuccessful in enrolling a matched control within their shift, interviewers requested for other interviewers to attempt enrollment in subsequent shifts via an instant messaging platform. To limit recall bias, cases were excluded in the primary analysis if not matched within 7 days. Interviewers documented the outcome (no answer, no consent, partial survey, completed survey) of each call and noted reasons for refusing participation or early call termination.

**Ethics and informed consent.** Verbal informed consent was obtained from all adult (aged ≥18 years) participants and parents/guardians of participants aged <18 years. The consenting parent/guardian was asked to answer on behalf of children aged <16 years however, they were able to invite children aged >7 years to participate in the interview if the child was willing, able, and interested. The informed consent script is available in S1 Item in S1 File. The State of California Health and Human Services Agency, Committee for the Protection of Human Subjects (Project 2021–034) approved the study protocol.

### Implementation infrastructure

Interviewers collected data daily (excluding holidays) for 10+ hours per week. Research associates, promoted from interviewers, helped maintain databases, manage interviewer training, assign call lists, facilitate weekly meetings, monitor enrollment, and cultivate community.

A communication platform provided live support to interviewers who encountered questions during surveys and served as an option for private and group communication. Supervisors monitored the platform daily to ensure timely response to questions. The platform streamlined communication to easily deliver critical updates, solicit feedback on survey implementation, and detect issues quickly.

The team met weekly to discuss enrollment progress, check-in on wellbeing, highlight interviewer accomplishments, and announce protocol or survey updates. Supervisors offered professional development opportunities during these meetings such as presentations from various public health professionals and workshops covering relevant skills and topics.

Interviewers intermittently encountered difficult conversations with participants. Team-wide, small-group, and 1-on-1 discussions about wellbeing recurred throughout the year to debrief difficult experiences, and mental wellness resources, including counseling and general support conferences across CDPH COVID-19 response sections, were advertised and encouraged.

## Interviewer team

Interviewers were recruited from undergraduate and graduate institutions with pay (S1 File). Successful candidates demonstrated strong empathy, patience, good communication, interest in public health or a related field, and had prior customer service, data collection, or healthcare experience. Interviewers completed a rigorous training program to ensure that they were well prepared for challenging interviews and collecting high-quality data (Fig 1). Due to high interviewer turnover in the first three months of the study, multiple hiring sessions occurred. With successive rounds of interviewer on-boarding we implemented a train-the-trainer approach, empowering experienced interviewers to mentor others and respond to questions.

## Quantitative methods

We define three cohorts representing different call outcomes: 1) individuals who answered the phone, 2) eligible individuals who consented, and 3) eligible individuals who refused participation in the study. To estimate determinants of participation, we estimated the adjusted odds ratio of answering the phone, consenting to participate, and citing time as a reason for not participating using mixed effects logistic regression. Models included age group, sex, region, SARS-CoV-2 infection status, month, time of day and time of week contacted as fixed effects and allowed random effects at the interviewer level. Additionally, we assessed interaction effects between predictors by SARS-CoV-2 infection status and between time of day and time of week. The Bayesian Information Criterion was used to compare models with and without interaction terms included (S1 File).

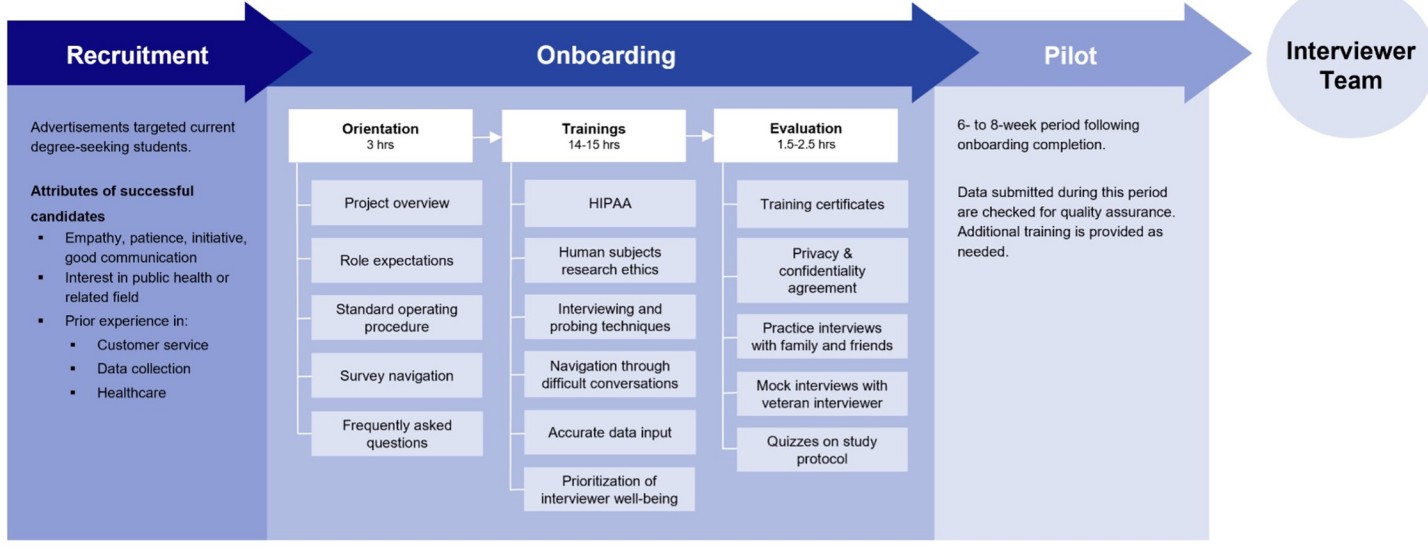

**Fig 1. Process diagram for recruitment, onboarding, and training interviewers.**

All analyses were conducted with R software (version 4.1.3; R Foundation for Statistical computing) and the lme4 package.

### Qualitative methods

From June 29 through July 12, 2022, we used an anonymous, self-administered, web-based survey to contextualize quantitative results with interviewer experiences (**S1 File**). All interviewers involved with the study were invited to participate. We compensated active interviewers for the time spent on their responses. We also reviewed weekly meeting notes for identification of themes.

## Results

During the study period, we placed 38,470 calls including 15,154 (39.4%) to cases and 23,316 (60.6%) to controls (**Table 1 and S1 File**). Among the cases and controls called, 35.5% (5,383/ 15,154) and 31.3% (7,289/23,316) answered the phone, respectively. Of those who answered the phone, 37.2% (2,004/5,383) and 27.3% (1,991/7,289) consented to participate. Ultimately, 1,885 cases and 1,871 controls completed the survey and were enrolled in the study. Over time, survey completion declined for both cases and controls despite change in calling rate (**Fig 2**).

On average, interviewers placed 8 calls per case and 13 calls per control to complete enrollment. Parents or guardians of children aged 0 to 4 years required fewer calls to complete enrollment compared to participants in other age categories (7 calls per case and 10 calls per control) (**S1 File**). The greatest number of calls to complete enrollment for a potential participant occurred between 8am to 11am (10 calls per case and 15 calls per control). On average, the weekly calls to complete enrollment for a control increased over time while remaining relatively steady for cases (**S1 File**).

During the study, three hiring rounds recruited a total of 34 interviewers. Interviewers were, on average, active for 23 weeks. 17.6% (6/34) remained active for 34–52 weeks. 32.4% (11/34) of interviewers responded to the anonymous experience survey.

### Predictors of answering the phone and consenting to participate

We found SARS-CoV-2 infection status, age, region, time of day called, and time of week called were significantly associated with answering the phone. Cases were more likely (aOR: 1.29 [95% CI: 1.24–1.35]) to answer the phone than controls (**Fig 3**). The likelihood of answering the phone was lowest among older individuals. Calls placed after 6pm (aOR: 0.79 [0.68, 0.90]) and between 8 to 11am (0.84 [0.79–0.90]) were associated with the lowest adjusted odds of answering the phone when compared to calls placed between 4 to 6pm (**S1 File**).

We also evaluated predictors of consenting to participate and found significant associations with SARS-CoV-2 infection status, age group, sex, and region. Cases had 1.69-fold higher adjusted odds of consenting compared to controls ([95%CI: 1.56–1.83], **Fig 3**). Women had 1.13-fold (1.04–1.22) higher adjusted odds of consenting than men. Parents or guardians of minors aged 0 to 4 were 1.53-times (1.18–1.98) more likely to consent than those aged 23 to 29.

Some motivations for participant consent, per interviewer reflections, were desire to contribute to public health research, relieve boredom, and express perspectives about the pandemic (**Table 2, Quotes 1–2**).

We identified differences in likelihood of consenting to participate occurred within SARS-CoV-2 infection status strata among age groups (aOR for cases 0.77 [95% CI: 0.62, 0.95] versus aOR for controls 1.14 [95% CI: 0.94,1.39] aged 60 and older) (**S1 File**).

**Table 1. Characteristics of SARS-CoV-2 test-seekers in California who were called, answered the phone, consented to participate, and completed the telephone survey.**

| | | Tested | Called | | Answered the phone[1] | | Consented to participate | | Completed the Survey | |
|---|---|---|---|---|---|---|---|---|---|---|
| | | n (%) | Case | Control | Case | Control | Case | Control | Case | Control |
| | | N = 81,980,132 | n (%) | n (%) | n (%) | n (%) | n (%) | n (%) | n (%) | n (%) |
| | | Case = 76428418 (93) Control = 5551714 (7) | N = 15154 | N = 23316 | N = 5081 | N = 6929 | N = 2004 | N = 1991 | N = 1885 | N = 1871 |
| Sex | Male | 37594032 (45.9) | 7410 (48.9) | 10969 (47.0) | 2496 (49.1) | 3357 (48.4) | 952 (47.5) | 954 (47.9) | 898 (47.6) | 896 (47.9) |
| | Female | 44386100 (54.1) | 7744 (51.1) | 12347 (53.0) | 2585 (50.9) | 3572 (51.6) | 1052 (52.5) | 1037 (52.1) | 987 (52.4) | 975 (52.1) |
| Age | 0 to 4 | 2357367 (2.9) | 359 (2.4) | 498 (2.1) | 130 (2.6) | 168 (2.4) | 64 (3.2) | 56 (2.8) | 59 (3.1) | 55 (2.9) |
| | 5 to 10 | 7186063 (8.8) | 664 (4.4) | 1062 (4.6) | 234 (4.6) | 357 (5.2) | 86 (4.3) | 109 (5.5) | 84 (4.5) | 98 (5.2) |
| | 11 to 13 | 3890275 (4.7) | 406 (2.7) | 595 (2.6) | 134 (2.6) | 181 (2.6) | 57 (2.8) | 50 (2.5) | 54 (2.9) | 47 (2.5) |
| | 14 to 17 | 5976574 (7.3) | 772 (5.1) | 1106 (4.7) | 250 (4.9) | 332 (4.8) | 92 (4.6) | 96 (4.8) | 86 (4.6) | 87 (4.6) |
| | 18 to 22 | 6799813 (8.3) | 1143 (7.5) | 2431 (10.4) | 413 (8.1) | 767 (11.1) | 195 (9.7) | 200 (10.0) | 176 (9.3) | 185 (9.9) |
| | 23 to 29 | 9942434 (12.1) | 2567 (16.9) | 4260 (18.3) | 904 (17.8) | 1317 (19.0) | 383 (19.1) | 377 (18.9) | 369 (19.6) | 356 (19.0) |
| | 30 to 39 | 13426157 (16.4) | 2830 (18.7) | 4381 (18.8) | 991 (19.5) | 1377 (19.9) | 376 (18.8) | 388 (19.5) | 358 (19.0) | 362 (19.3) |
| | 40 to 49 | 10816874 (13.2) | 2230 (14.7) | 3396 (14.6) | 755 (14.9) | 996 (14.4) | 299 (14.9) | 288 (14.5) | 276 (14.6) | 277 (14.8) |
| | 50 to 59 | 9699282 (11.8) | 1798 (11.9) | 2513 (10.8) | 589 (11.6) | 636 (9.2) | 211 (10.5) | 176 (8.8) | 200 (10.6) | 168 (9.0) |
| | 60+ | 11885293 (14.5) | 2385 (15.7) | 3074 (13.2) | 681 (13.4) | 798 (11.5) | 241 (12.0) | 251 (12.6) | 223 (11.8) | 236 (12.6) |
| Region | San Francisco Bay Area | 18078093 (22.1) | 1432 (9.4) | 2317 (9.9) | 539 (10.6) | 758 (10.9) | 231 (11.5) | 228 (11.5) | 218 (11.6) | 212 (11.3) |
| | Central Coast | 626715 (0.8) | 1642 (10.8) | 2880 (12.4) | 557 (11.0) | 742 (10.7) | 242 (12.1) | 225 (11.3) | 223 (11.8) | 218 (11.7) |
| | Greater Sacramento Area | 2686924 (3.3) | 1617 (10.7) | 2618 (11.2) | 553 (10.9) | 867 (12.5) | 236 (11.8) | 241 (12.1) | 221 (11.7) | 225 (12.0) |
| | Northern Sacramento Valley | 1762248 (2.1) | 1442 (9.5) | 2179 (9.3) | 472 (9.3) | 712 (10.3) | 208 (10.4) | 208 (10.4) | 197 (10.5) | 199 (10.6) |
| | San Joaquin Valley | 6707663 (8.2) | 1957 (12.9) | 3174 (13.6) | 664 (13.1) | 917 (13.2) | 239 (11.9) | 232 (11.7) | 220 (11.7) | 219 (11.7) |
| | Northwestern California | 840819 (1.0) | 1540 (10.2) | 2203 (9.4) | 493 (9.7) | 617 (8.9) | 213 (10.6) | 213 (10.7) | 201 (10.7) | 198 (10.6) |
| | Sierras | 1465520 (1.8) | 1655 (10.9) | 2401 (10.3) | 529 (10.4) | 724 (10.4) | 198 (9.9) | 208 (10.4) | 189 (10.0) | 190 (10.2) |
| | San Diego and southern border | 6692230 (8.2) | 1673 (11.0) | 2617 (11.2) | 591 (11.6) | 747 (10.8) | 214 (10.7) | 219 (11.0) | 203 (10.8) | 206 (11.0) |
| | Greater Los Angeles Area | 43119920 (52.6) | 2196 (14.5) | 2927 (12.6) | 683 (13.4) | 845 (12.2) | 223 (11.1) | 217 (10.9) | 213 (11.3) | 204 (10.9) |
| Month | February (2021) | 4937436 (6.0) | 177 (1.2) | 228 (1.0) | 66 (1.3) | 78 (1.1) | 35 (1.7) | 30 (1.5) | 34 (1.8) | 29 (1.5) |
| | March | 4675127 (5.7) | 1694 (11.2) | 2443 (10.5) | 651 (12.8) | 844 (12.2) | 301 (15.0) | 298 (15.0) | 272 (14.4) | 279 (14.9) |
| | April | 4708939 (5.7) | 2009 (13.3) | 2661 (11.4) | 760 (15.0) | 869 (12.5) | 324 (16.2) | 309 (15.5) | 302 (16.0) | 289 (15.4) |
| | May | 4051343 (4.9) | 1539 (10.2) | 2328 (10.0) | 569 (11.2) | 754 (10.9) | 224 (11.2) | 225 (11.3) | 209 (11.1) | 206 (11.0) |
| | June | 3150746 (3.8) | 1346 (8.9) | 1683 (7.2) | 428 (8.4) | 516 (7.4) | 158 (7.9) | 156 (7.8) | 149 (7.9) | 146 (7.8) |
| | July | 3474680 (4.2) | 1040 (6.9) | 1487 (6.4) | 350 (6.9) | 450 (6.5) | 146 (7.3) | 138 (6.9) | 137 (7.3) | 133 (7.1) |
| | August | 6679276 (8.1) | 744 (4.9) | 1207 (5.2) | 271 (5.3) | 389 (5.6) | 116 (5.8) | 118 (5.9) | 111 (5.9) | 110 (5.9) |
| | September | 7596972 (9.3) | 600 (4.0) | 987 (4.2) | 205 (4.0) | 324 (4.7) | 87 (4.3) | 92 (4.6) | 82 (4.4) | 85 (4.5) |
| | October | 6647015 (8.1) | 924 (6.1) | 1884 (8.1) | 319 (6.3) | 563 (8.1) | 110 (5.5) | 119 (6.0) | 108 (5.7) | 110 (5.9) |
| | November | 5748797 (7.0) | 1098 (7.2) | 1662 (7.1) | 345 (6.8) | 457 (6.6) | 106 (5.3) | 98 (4.9) | 100 (5.3) | 97 (5.2) |
| | December | 7875324 (9.6) | 1344 (8.9) | 2816 (12.1) | 394 (7.8) | 682 (9.8) | 154 (7.7) | 161 (8.1) | 146 (7.7) | 149 (8.0) |
| | January (2022) | 14487871 (18) | 1665 (11.0) | 2488 (10.7) | 467 (9.2) | 666 (9.6) | 171 (8.5) | 171 (8.6) | 166 (8.8) | 168 (9.0) |
| | February | 7946606 (10) | 974 (6.4) | 1442 (6.2) | 256 (5.0) | 337 (4.9) | 72 (3.6) | 76 (3.8) | 69 (3.7) | 70 (3.7) |

(*Continued*)

**Table 1.** (*Continued*)

| | | Tested | Called | | Answered the phone[1] | | Consented to participate | | Completed the Survey | |
|---|---|---|---|---|---|---|---|---|---|---|
| | | *n* (%) | Case | Control | Case | Control | Case | Control | Case | Control |
| | | *N* = 81,980,132 | *n* (%) | *n* (%) | *n* (%) | *n* (%) | *n* (%) | *n* (%) | *n* (%) | *n* (%) |
| | | Case = 76428418 (93) Control = 5551714 (7) | *N* = 15154 | *N* = 23316 | *N* = 5081 | *N* = 6929 | *N* = 2004 | *N* = 1991 | *N* = 1885 | *N* = 1871 |
| Time of week | Weekday | | 12384 (81.7) | 19115 (82.0) | 4138 (81.4) | 5725 (82.6) | 1656 (82.6) | 1641 (82.4) | 1561 (82.8) | 1539 (82.3) |
| | Weekend | | 2770 (18.3) | 4201 (18.0) | 943 (18.6) | 1204 (17.4) | 348 (17.4) | 350 (17.6) | 324 (17.2) | 332 (17.7) |
| Time of day | 8-11am | | 4272 (28.2) | 4833 (20.7) | 1355 (26.7) | 1336 (19.3) | 494 (24.7) | 347 (17.4) | 455 (24.1) | 331 (17.7) |
| | 12-3pm | | 7090 (46.8) | 11792 (50.6) | 2428 (47.8) | 3475 (50.2) | 996 (49.7) | 989 (49.7) | 945 (50.1) | 924 (49.4) |
| | 4-6pm | | 3483 (23.0) | 5798 (24.9) | 1198 (23.6) | 1855 (26.8) | 465 (23.2) | 563 (28.3) | 441 (23.4) | 532 (28.4) |
| | After 6pm | | 309 (2.0) | 893 (3.8) | 100 (2.0) | 263 (3.8) | 49 (2.4) | 92 (4.6) | 44 (2.3) | 84 (4.5) |

See **S8 Table in S1 File** for comparison to 2020 U.S. Census Bureau American Community Survey demographics.

[1]This is restricted to the subpopulation of individuals who answered the phone and were eligible for the study. The 662 potential participants (360 controls and 302 cases) who answered the telephone call but were ineligible for the study and therefore excluded from this count (**Fig 1**).

## Reasons for refusing participation

We identified differences in the reasons for refusing participation among 8,015 eligible individuals who answered the phone. The majority (90.9%; 7285/8015) cited insufficient time as the reason for refusing participation, with the proportion citing this reason increasing over time (**S1 File**). Others cited language barriers (2.6%; 206/8015), lack of interest (2.3%; 184/8015), call fatigue (0.45%; 36/8015), and/or being unwell or grieving (0.45%; 36/8015) as reasons (**S1 File**).

We assessed determinants of indicating insufficient time as a reason for refusing participation. Cases were associated with a 0.44-fold (95% CI: 0.37–0.53) lower adjusted odds of citing insufficient time compared to controls (**S1 File**). Individuals aged 23 to 29 were most likely to cite insufficient time compared to all other age categories. We did not find evidence of significant associations between the time of day or week the individual was called and citing time as a reason for refusing participation.

Although interviewers observed that individuals often refused based on timing, they identified additional reasons, including personal beliefs, distrust, illness, and stress (**Table 2, Quotes 3–4**).

## Sample diversity

Participants completing the survey were comparable to the SARS-CoV-2 test seeking population in California across sex and in age groups 0–4, 18–22, 40–49, and 60+ (**Table 1 and S1 File**). By design, participants were enrolled equally across each study region. The composition of study participants was roughly proportional to the state by household income and race/ethnicity (**S7 Fig**).

Pandemic sentiments and behaviors, self-reported by participants, were diverse and changed over time. Agreement with social distancing and face mask recommendations generally remained constant throughout the study period (**S1 File**), however, anxiety about the pandemic fluctuated between 67.7% in February 2021 (44/65 participants) and 29.6% in July 2021

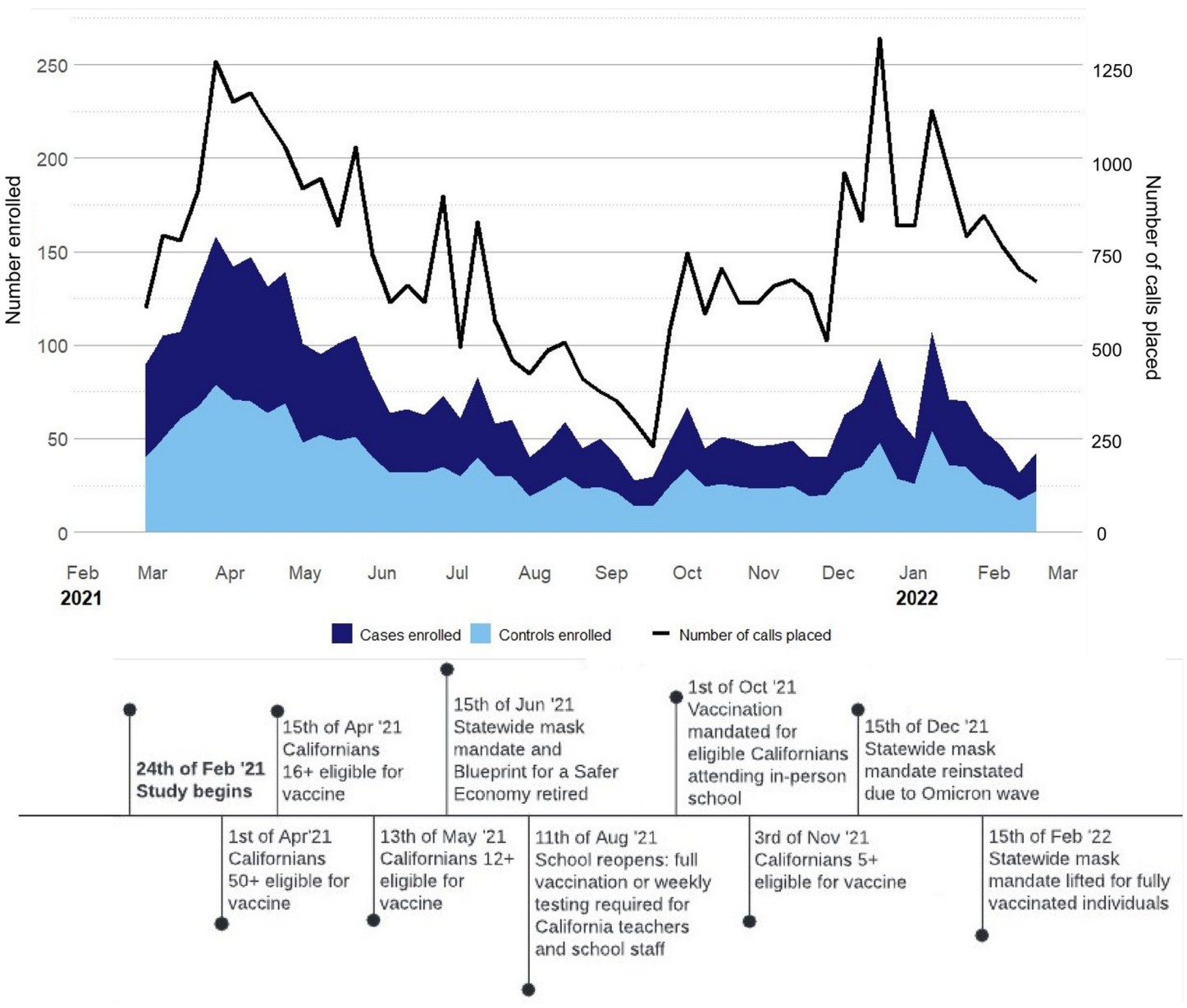

**Fig 2. Study timeline mapped against weekly enrollment trends by case-control status.**

(84/284 participants). Participants reporting visiting two or more public indoor settings within the two weeks prior to getting tested increased from 58.5% (38/65) in February 2021 to 85.1% (126/148) by February 2022. Attendance to each type of indoor setting remained constant - except for a decrease in grocery store visits and increase in school visits (**S1 File**). The proportion of individuals ineligible for enrollment due to previously being infected with SARS-CoV-2 increased throughout the study period (**S1 File**).

Emotional states among participants, as encountered by interviewers, were also variable. The range of pandemic-related emotions that participants expressed included resilience, weariness, loneliness, and anger (**Table 2, Quotes 5–6**).

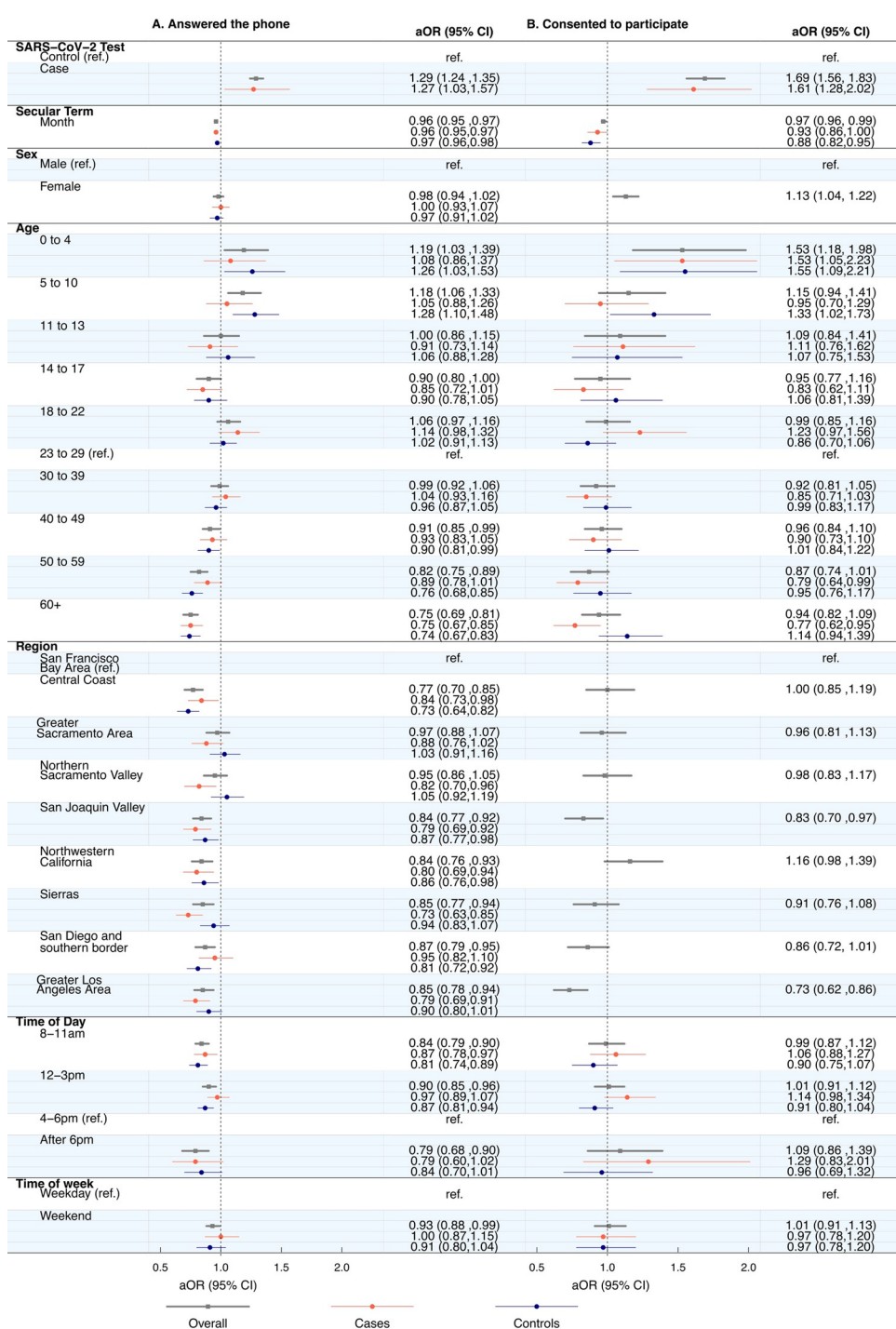

**Fig 3. Predictors of participants answering the telephone and consenting to participate in the California COVID-19 Case Control study.** We did not observe significant interaction between SARS-CoV-2 infection status and the adjusted odds of consenting to participate by region or sex. Estimates for cases and controls are not pictured for these two predictors of consenting to participate.

Some interviewers reported occurrences of previously vaccine-opposed participants expressing willingness to seek COVID-19 vaccination after testing positive (**Table 2, Quote** 7).

**Table 2. Interviewer experience survey quotes.**

| | Section | Quote |
|---|---|---|
| 1 | Predictors of answering the phone and consenting to participate | *Participants genuinely believe their answers will help end the pandemic in some way.* |
| 2 | Predictors of answering the phone and consenting to participate | *I think for the people who were inclined to not participate because of beliefs, some changed their minds when it was reframed as us wanting to make sure everyone is represented, that their voice matters, and this is a chance for them to be heard.* |
| 3 | Reasons for refusing to participate | *Some people that were politically against the public health response often declined to get interviewed or were outright confrontational.* |
| 4 | Reasons for refusing to participate | *It is stressful for COVID-19 positive and negative cases to follow through with the interview due to sickness, worry, or even suspicions of the intents and validity of our study.* |
| 5 | Sample diversity | *A lot of people stated that they were so tired of living through a pandemic.* |
| 6 | Sample diversity | *Some people angrily shared experiences about being knowingly exposed to COVID-19 by their bosses or clients during work.* |
| 7 | Sample diversity | *A participant who was firmly against believing in covid ended up changing his mind after testing positive. He told me that he would get the vaccine when he could.* |
| 8 | Impact on interviewer wellbeing | *Calls that I had where people shared their anxiety, confusion, fear, anger, and sadness fed into my own anxiety and negative feelings.* |
| 9 | Impact on interviewer wellbeing | *A lot of people said, "thank you for what you're doing." . . . That made me proud to be part of such an important research group.* |
| 10 | Structural successes and adaptations | *We were allowed to give feedback (and our feedback was valuable and used in survey changes), and supervisors cared about our mental health over collecting data.* |
| 11 | Structural successes and adaptations | *The team was very supportive when I was sharing my experience, which helped show me that it is normal to feel the impact the participants may have on us whether they are at their highest or their lowest.* |
| 12 | Structural successes and adaptations | *Seeing the data I had helped collect be used in real time to help improve understandings of COVID inspired me to keep calling people, even when I would reach voicemail after voicemail.* |
| 13 | Structural successes and adaptations | *Because it was a remote job, I sometimes felt as though I was working alone, but weekly meetings helped provide that sense of teamwork.* |

## Interviewer wellbeing

Of the 11 interviewers who responded to the interviewer experience survey, 63.6% (7) stated they occasionally encountered scenarios where they were compelled to search for or connect participants to social services and 18.3% (2) stated they encountered this need often (**S1 File**). Resources pertaining to healthcare access (COVID-19 or general), housing security, food security, and financial relief were the most frequently requested. Most interviewers reported encountering participant grief or anger occasionally (81.8%, 9/11). 18.2% (2) reported encountering anger often.

Interviewers reported poor mental health and lingering feelings after difficult calls when participants discussed socioeconomic burdens, pandemic hardship, grief, suffering, inequitable conditions, or acted with hostility and bullying. Interviewers felt stressed, especially when participants compelled them to fulfill social service or counselor roles (**Table 2**, **Quote 8**).

Interviewers also described many encounters which instilled a sense of purpose, pride, spurred personal growth, cultivated a sense of community, expanded empathy, and uplifted

moods. Notably, encounters when participants expressed appreciation, gratitude, humor, or warmth despite hardships had resounding effects on interviewers (**Table 2, Quote 9**). Interviewers also mentioned how the study provided remote career growth and employment during a time of scarce opportunities.

## Structural successes and adaptations

Feedback was frequently solicited to identify improvement opportunities. When mental health concerns surfaced, quick action was taken to strengthen structural support, community engagement, and resources. Research associates, with experience as interviewers, developed and led robust training that emphasized mental wellbeing and methods to navigate difficult conversations. They also compiled information on frequently requested social services and expanded on the standard operating procedure with scenario-specific protocols and responses to demands beyond interviewer duties. Active efforts to sustain a work environment that felt safe, supportive, and caring were made to better protect the mental health of interviewers (**Table 2, Quote 10**).

Interviewers reported certain structural components as being particularly beneficial: self-assigned scheduling of shift times and weekly meetings. Self-assigned shifts allowed interviewers affected by difficult conversations to take breaks. Meeting weekly helped boost team morale, relieve isolation, and created bonding between team members (**Table 2, Quotes 11–13**).

## Discussion

Over a one-year period during the COVID-19 pandemic, 9.8% of 38,470 individuals invited to our phone-based questionnaire consented to participate. Because the study was conducted across an evolving landscape of COVID-19 epidemiology and public health recommendations, flexibility to adapt protocols, exclusion criteria, and survey questions so that they remained meaningful was necessary. Results were consistent with prior research demonstrating that individuals who have a history of disease are more willing to participate in a health study than those naïve to the disease [11]. The likelihood of an individual answering the phone decreased with age. Older individuals may experience more severe health burdens or reside in institutions unreachable by direct calls [18,24]. The time of day that a potential participant was called influenced the likelihood of answering the phone, but not of consenting to participate. Results confirmed literature reporting that morning calls yield lower enrollment, indicating that strategically timing calls is crucial in maximizing enrollment efficiency [15]. We recommend placing calls during the afternoon and evening, allocating more efforts towards enrolling controls, and restricting survey length if possible.

This study was successful in representing the population seeking SARS-CoV-2 testing in California, with a recruitment effort of almost 40,000 calls and a well-powered size of nearly 4,000 participants within the first year. The data quality allowed for identification of reasons for unsuccessful enrollment and determinants of participation. The infrastructure of the study, particularly weekly meetings, detailed standard operating procedure documentation, and messaging platform enabled quick identification of obstacles and implementation adaptations.

Enrollment—especially of controls—became more difficult throughout the study. This may be explained by the increase in previously positive individuals and by diminished interest or perceived risk regarding the pandemic. We recommend shortening survey length or offering call-backs to minimize loss of participants due to insufficient time.

Interviewers highlighted themes unique to remote phone-based research during the COVID-19 pandemic. Notably, strong participant emotions and harassment were especially trying for some interviewers. We believe these findings are novel in remote, phone-based

quantitative health research and unique to the national context of polarized attitudes towards the COVID-19 pandemic [25]. Proactively adapting to emerging obstacles was critical to the success of the study. Designing training that simulated realistic scenarios and detailed protocols for difficult encounters resulted in considerable improvement for subsequent interviewer cohorts. We suggest implementation of frequent proactive mental health check-ins, continual collection of anonymous feedback, and an exit survey for interviewers.

There are several limitations to this analysis. Due to data constraints, we were unable to examine how socioeconomic status, race, education, occupation, and setting, such as housing, may influence the likelihood of answering the phone and consenting to participate. Results may not be generalizable to the broader California population, as individuals who did not seek laboratory-confirmed SARS-CoV-2 testing are excluded by design. Severely ill SARS-CoV-2 positive individuals, unwell individuals with comorbidities, those without stable phone service, and those cautious about phone solicitations might not be well represented in our study.

## Conclusions

Our findings demonstrate how researchers can strategize recruitment for future phone-based observational studies conducted amidst an evolving public health emergency. Actively monitoring study implementation enables timely adaptation of practices for data collection and can be an important approach to preserving interviewer and other study staff well-being. We provide evidence of poor mental health and burnout among remote study staff that is consistent with previous literature on public health workers. Our findings will assist future researchers in conducting efficient, sustainable, and timely research in response to emergent public health crises.

## Supporting information

**S1 File. Supplementary materials.**
(DOCX)

## Acknowledgments

We would like to thank all study participants that gave time to complete our survey making possible this work.

Members of the California COVID-19 Case-Control Study Team include: Adrian Cornejo, Amanda Lam, Amanda Moe, Amandeep Kaur, Anna Fang, Ashly Dyke, Camilla Barbaduomo, Christine Wan, Diana Nicole Morales Felipe, Diana Poindexter, Erin Xavier, Hyemin Park, Helia Samani, Jessica Ni, Julia Cheunkarndee, Mahsa Javadi, Maya Spencer, Michelle Spinosa, Miriam Bermejo, Monique Miller, Najla Dabbagh, Natalie Dassian, Nikolina Walas, Paulina Frost, Savannah Corredor, Shrey Saretha, Timothy Ho, Vivian Tran, Yang Zhou, Yasmine Abdulrahim, Zheng Dong.

## Author Contributions

**Conceptualization:** John Openshaw, James Watt, Joseph A. Lewnard, Seema Jain, Kristin L. Andrejko, Jake M. Pry.

**Data curation:** Nozomi Fukui, Sophia S. Li, Jennifer DeGuzman, Jennifer F. Myers.

**Formal analysis:** Nozomi Fukui, Sophia S. Li, Anjali Sharma, Joseph A. Lewnard, Jake M. Pry.

**Funding acquisition:** Seema Jain, Jake M. Pry.

**Investigation:** Nozomi Fukui, Sophia S. Li, Joseph A. Lewnard, Kristin L. Andrejko.

**Methodology:** Sophia S. Li, Anjali Sharma, Joseph A. Lewnard, Kristin L. Andrejko, Jake M. Pry.

**Project administration:** Jennifer DeGuzman, Kristin L. Andrejko, Jake M. Pry.

**Resources:** Jake M. Pry.

**Supervision:** John Openshaw, Joseph A. Lewnard, Seema Jain, Kristin L. Andrejko, Jake M. Pry.

**Validation:** Jennifer F. Myers, Jake M. Pry.

**Visualization:** Nozomi Fukui, Sophia S. Li.

**Writing – original draft:** Nozomi Fukui, Sophia S. Li.

**Writing – review & editing:** Nozomi Fukui, Sophia S. Li, Anjali Sharma, James Watt, Joseph A. Lewnard, Seema Jain, Kristin L. Andrejko, Jake M. Pry.

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
