## [Decision Letter · Decision Letter 0]

19 Feb 2024

PONE-D-23-40982Mixed methods approach to examining the implementation experience of a phone-based health research survey investigating risk factors for SARS-CoV-2 infection in CaliforniaPLOS ONE

Dear Dr. Pry,

Thank you for submitting your manuscript to PLOS ONE. After careful consideration, we feel that it has merit but does not fully meet PLOS ONE’s publication criteria as it currently stands. Therefore, we invite you to submit a revised version of the manuscript that addresses the points raised during the review process.

We look forward to receiving your revised manuscript.

Kind regards,

Moustaq Karim Khan Rony

Academic Editor

PLOS ONE

Journal Requirements:

"This study was supported by the Centers for Disease Control and Prevention, Enhanced  Epidemiology and Laboratory Capacity (ELC) grant number: 5-NU50CK000539."

4. Thank you for stating the following in your Competing Interests section: "NA"

5. In the online submission form, you indicated that "De-identified data may be made available upon to the corresponding author with approval from the California Department of Public Health Office of Human Subjects Protection."

6. One of the noted authors is a group or consortium [California COVID-19 Case-Control Study Team]. In addition to naming the author group, please list the individual authors and affiliations within this group in the acknowledgments section of your manuscript. Please also indicate clearly a lead author for this group along with a contact email address.’ 

7. Your ethics statement should only appear in the Methods section of your manuscript. If your ethics statement is written in any section besides the Methods, please move it to the Methods section and delete it from any other section. Please ensure that your ethics statement is included in your manuscript, as the ethics statement entered into the online submission form will not be published alongside your manuscript.

**Additional Editor Comments:**

This study is very interesting, and no major issues were found. Please consider the reviewer comments below, and any changes for improvement will be appreciated.

Reviewers' comments:

Reviewer's Responses to Questions

**Comments to the Author**

1. Is the manuscript technically sound, and do the data support the conclusions?

Reviewer #1: Yes

2. Has the statistical analysis been performed appropriately and rigorously? 

Reviewer #1: Yes

3. Have the authors made all data underlying the findings in their manuscript fully available?

Reviewer #1: Yes

4. Is the manuscript presented in an intelligible fashion and written in standard English?

Reviewer #1: Yes

5. Review Comments to the Author

Reviewer #1: In their manuscript, Fukui and colleagues implemented a test-negative case-control study including participants residing in California to inform the design and implementation strategies of observational research in pandemic settings. Results showed that patients with COVID-19 had a higher odds of answering the phone and consenting to participate in the study, compared to control subjects (COVID-19-negative participants). Also, calls placed from 4pm to 6pm had the highest adjusted odds of being answered. The authors also highlighted aspects of interacting with participants with physical and emotional needs, and documented verbal harassment cases from individuals called.

Although the topic may not currently be up-to-date, the study is appropriately designed and well-written. Consider the following for improvement:

1. Lines 1-3: The manuscript mainly focused on identifying factors that might influence participation in a phone-based research survey. The candidate pool was based on a survey examining risk factors for COVID-19 infection. However, the way the title is phrased implies that risk factors for COVID-19 were actively researched in this paper. Please, revise the title to accurately describe the purpose of the study.

2. Lines 109-110: Based on supplementary figures S2 and S3, a laboratory test was used for determining eligibility for the study. Why is clinical diagnosis of COVID-19 described as an eligibility criterion in the Methods? Were eligible participants recruited based on lab results, clinical symptomatology, or both?

3. The results showed that individuals with COVID-19 had a higher odds of answering the phone and consenting to participate in the study. From the supplementary materials, it seems that participants were asked to comment on their symptoms, if any. Did the authors run a separate analysis comparing patients with COVID-19 who were asymptomatic vs. symptomatic? If no, could such an approach be explored as well? It might be just a speculation, but I think that symptomatic patients would be more willing to participate.

6. PLOS authors have the option to publish the peer review history of their article (what does this mean?). If published, this will include your full peer review and any attached files.

Reviewer #1: No

---

## [Author Response · Author response to Decision Letter 0]

8 Mar 2024

Dr. Moustaq Karim Khan Rony 

Academic Editor

PLOS One

6 March 2024

Dear Dr. Karim Khan Rony,

Many thanks for re-reviewing our revised manuscript (PONE-D-23-40982-R1), attached, entitled: “Mixed methods approach to examining the implementation experience of a phone-based health research survey investigating risk factors for SARS-CoV-2 infection in California”. 

On behalf of my colleagues, I would like to thank you and the reviewers for the thoughtful review. We have considered each comment carefully and have made the changes noted below, which we believe have again improved the manuscript. We have enclosed two versions of the revised manuscript in Word (.docx)—one with changes highlighted under “track changes” and a second “clean” version with all changes accepted. 

Please find here our point-by-point responses to reviewer comments in green font, organized by headings taken from the reviewers provided on 19 February 2024 (also, please note that for ease of reference the lines below refer to the “clean” version of the revised manuscript):

Begin itemized review response.

Editor(s)' Comments to Author (if any):

Journal Requirements:

"This study was supported by the Centers for Disease Control and Prevention, Enhanced Epidemiology and Laboratory Capacity (ELC) grant number: 5-NU50CK000539."

Author response: The following statement is correct "The funders had no role in study design, data collection and analysis, decision to publish, or preparation of the manuscript." Many thanks for seeing to the amendment.

4. Thank you for stating the following in your Competing Interests section: "NA"

Author response: Thank you for offering to change the submission form on our behalf. Co-author J.A.L. discloses the following financial interests/personal relationships which may be considered as potential competing interests: receipt of grants and honoraria from Pfizer, Inc, outside the submitted work. All other authors have declared that no competing interests exist.

5. In the online submission form, you indicated that "De-identified data may be made available upon to the corresponding author with approval from the California Department of Public Health Office of Human Subjects Protection."

Author response: We have uploaded the deidentified data to Data Dryad for public access and provided the link in the manuscript under “Data Availability” subsection.

6. One of the noted authors is a group or consortium [California COVID-19 Case-Control Study Team]. In addition to naming the author group, please list the individual authors and affiliations within this group in the acknowledgments section of your manuscript. Please also indicate clearly a lead author for this group along with a contact email address.’ 

Author response: Thank you for this note, we have added authors in that group to the acknowledgements section.

7. Your ethics statement should only appear in the Methods section of your manuscript. If your ethics statement is written in any section besides the Methods, please move it to the Methods section and delete it from any other section. Please ensure that your ethics statement is included in your manuscript, as the ethics statement entered into the online submission form will not be published alongside your manuscript.

Author response: Thank you, we have revised the manuscript accordingly.

Author response: Thank you for this guidance. We have amended in text citations and added captions for the Supporting Information files at the end of the manuscript.

Author response: Thank you, we have reviewed references and adjusted citations per guidance.

Additional Editor Comments:

This study is very interesting, and no major issues were found. Please consider the reviewer comments below, and any changes for improvement will be appreciated.

Author response: many thanks for sharing your thoughts, we appreciate the positive feedback.

Reviewers' comments:

Reviewer's Responses to Questions

Comments to the Author

1. Is the manuscript technically sound, and do the data support the conclusions?

Reviewer #1: Yes

2. Has the statistical analysis been performed appropriately and rigorously? 

Reviewer #1: Yes

3. Have the authors made all data underlying the findings in their manuscript fully available?

Reviewer #1: Yes

4. Is the manuscript presented in an intelligible fashion and written in standard English?

Reviewer #1: Yes

5. Review Comments to the Author

Reviewer #1: In their manuscript, Fukui and colleagues implemented a test-negative case-control study including participants residing in California to inform the design and implementation strategies of observational research in pandemic settings. Results showed that patients with COVID-19 had a higher odds of answering the phone and consenting to participate in the study, compared to control subjects (COVID-19-negative participants). Also, calls placed from 4pm to 6pm had the highest adjusted odds of being answered. The authors also highlighted aspects of interacting with participants with physical and emotional needs, and documented verbal harassment cases from individuals called.

Although the topic may not currently be up-to-date, the study is appropriately designed and well-written. Consider the following for improvement:

1. Lines 1-3: The manuscript mainly focused on identifying factors that might influence participation in a phone-based research survey. The candidate pool was based on a survey examining risk factors for COVID-19 infection. However, the way the title is phrased implies that risk factors for COVID-19 were actively researched in this paper. Please, revise the title to accurately describe the purpose of the study.

Author response: We have revised the title to better reflect this manuscript. It is now titled “Mixed methods approach to examining the implementation experience of a phone-based survey for a SARS-CoV-2 test-negative case-control study in California”.

2. Lines 109-110: Based on supplementary figures S2 and S3, a laboratory test was used for determining eligibility for the study. Why is clinical diagnosis of COVID-19 described as an eligibility criterion in the Methods? Were eligible participants recruited based on lab results, clinical symptomatology, or both?

Author response: The potential participant’s most recent SARS-CoV-2 laboratory test was obtained from case and control linelists generated from all mandatory reporting to the state surveillance system (CalREDIE) within the past 72 hours. Once a case or a control that was matched to an enrolled case on the linelist was called, they were asked this additional question about prior COVID-19 diagnosis or prior positive test results (not including their most recent test that we generating the line lists from). We wanted to ensure that participants, both cases and controls, had no prior infection of COVID-19 before enrolling them in the study. 

3. The results showed that individuals with COVID-19 had a higher odds of answering the phone and consenting to participate in the study. From the supplementary materials, it seems that participants were asked to comment on their symptoms, if any. Did the authors run a separate analysis comparing patients with COVID-19 who were asymptomatic vs. symptomatic? If no, could such an approach be explored as well? It might be just a speculation, but I think that symptomatic patients would be more willing to participate.

Author response: Thank you for this suggestion. We did not run this analysis however, we acknowledge the limitation that patients with milder symptoms may be more willing to participate, while those who were more severely ill might have been unreachable, too unwell to give informed consent, or less willing to participate.

End itemized review response.

We would, again, like to thank the editor and reviewers for their time and thoughtful comments. We believe that edits made in response to the reviewer notes have further strengthened the manuscript and we look forward to receiving your feedback in due course.

Kind Regards,

Jake M. Pry, PhD, MPH

Assistant Professor

Department of Public Health Sciences, University of California, Davis

jmpry@ucdavis.edu | jake.pry@cidrz.org

---

## [Decision Letter · Decision Letter 1]

11 Mar 2024

Mixed methods approach to examining the implementation experience of a phone-based survey for a SARS-CoV-2 test-negative case-control study in California

PONE-D-23-40982R1

Dear Dr. Pry,

We’re pleased to inform you that your manuscript has been judged scientifically suitable for publication and will be formally accepted for publication once it meets all outstanding technical requirements.

Kind regards,

Moustaq Karim Khan Rony

Academic Editor

PLOS ONE

Additional Editor Comments (optional):

Reviewers' comments:

Reviewer's Responses to Questions

**Comments to the Author**

1. If the authors have adequately addressed your comments raised in a previous round of review and you feel that this manuscript is now acceptable for publication, you may indicate that here to bypass the “Comments to the Author” section, enter your conflict of interest statement in the “Confidential to Editor” section, and submit your "Accept" recommendation.

Reviewer #1: All comments have been addressed

2. Is the manuscript technically sound, and do the data support the conclusions?

Reviewer #1: Yes

3. Has the statistical analysis been performed appropriately and rigorously? 

Reviewer #1: Yes

4. Have the authors made all data underlying the findings in their manuscript fully available?

Reviewer #1: Yes

5. Is the manuscript presented in an intelligible fashion and written in standard English?

Reviewer #1: Yes

6. Review Comments to the Author

Reviewer #1: The authors have addressed my concerns. No further comments.

I endorse the publication of this manuscript.

7. PLOS authors have the option to publish the peer review history of their article (what does this mean?). If published, this will include your full peer review and any attached files.

Reviewer #1: No

---

## [Editor Report · Acceptance letter]

30 Apr 2024

PONE-D-23-40982R1 

PLOS ONE

Dear Dr. Pry, 

I'm pleased to inform you that your manuscript has been deemed suitable for publication in PLOS ONE. Congratulations! Your manuscript is now being handed over to our production team.

Kind regards, 

on behalf of

Dr. Moustaq Karim Khan Rony 

Academic Editor

PLOS ONE